# DeepCME: A deep learning framework for computing solution statistics of the chemical master equation

**Ankit Gupta**[1], **Christoph Schwab**[2], **Mustafa Khammash**[1]*

**1** Department of Biosystems Science and Engineering, ETH Zürich, Basel, Switzerland, **2** Seminar für Angewandte Mathematik, ETH Zürich, Zürich, Switzerland

* mustafa.khammash@bsse.ethz.ch

**Data Availability Statement:** The source code used to produce the results and analyses presented in this manuscript are available from

## Abstract

Stochastic models of biomolecular reaction networks are commonly employed in systems and synthetic biology to study the effects of stochastic fluctuations emanating from reactions involving species with low copy-numbers. For such models, the Kolmogorov's forward equation is called the chemical master equation (CME), and it is a fundamental system of linear ordinary differential equations (ODEs) that describes the evolution of the probability distribution of the random state-vector representing the copy-numbers of all the reacting species. The size of this system is given by the number of states that are accessible by the chemical system, and for most examples of interest this number is either very large or infinite. Moreover, approximations that reduce the size of the system by retaining only a finite number of important chemical states (e.g. those with non-negligible probability) result in high-dimensional ODE systems, even when the number of reacting species is small. Consequently, accurate numerical solution of the CME is very challenging, despite the linear nature of the underlying ODEs. One often resorts to estimating the solutions via computationally intensive stochastic simulations. The goal of the present paper is to develop a novel deep-learning approach for computing solution statistics of high-dimensional CMEs by reformulating the stochastic dynamics using Kolmogorov's backward equation. The proposed method leverages superior approximation properties of Deep Neural Networks (DNNs) to reliably estimate expectations under the CME solution for several user-defined functions of the state-vector. This method is algorithmically based on reinforcement learning and it only requires a moderate number of stochastic simulations (in comparison to typical simulation-based approaches) to train the "policy function". This allows not just the numerical approximation of various expectations for the CME solution but also of its sensitivities with respect to all the reaction network parameters (e.g. rate constants). We provide four examples to illustrate our methodology and provide several directions for future research.

GitHub repository: https://github.com/ankitgupta83/DeepCME.

**Funding:** This project has received funding from the European Research Council (ERC) https://erc.europa.eu/ under the European Union's Horizon 2020 research and innovation programme grant agreement no. 743269 (CyberGenetics project). This grant was awarded to M.K. The funders had no role in study design, data collection and analysis, decision to publish, or preparation of the manuscript.

**Competing interests:** The authors have declared that no competing interests exist.

## Author summary

We develop a deep learning framework for estimating solutions of the chemical master equation (CME) that is fundamental to stochastic analysis of reaction networks. The CME is a system of ordinary differential equations that describes the time-evolution of the probability density of the random state-vector, and owing to an inherent curse of dimensionality, directly solving the CME is generally impractical with existing approaches. Moreover, the commonly employed simulation-based approaches for estimating CME solutions often require an exorbitant amount of computational time, even for moderately-sized networks. To counter these issues, we develop a deep reinforcement learning based method, called *DeepCME*, in this paper. DeepCME not only estimates function expectations based on the CME solution, but it also solves the more challenging problem of estimating their sensitivities with respect to all the model parameters. We illustrate our approach with four carefully chosen reaction network examples with varying sizes. Our results demonstrate that DeepCME reliably estimates the expectations of interest, along with all the parametric sensitivities, at a fraction of the computational cost of simulation-based estimators. We present many directions for future research and suggest further improvements to DeepCME that can greatly enhance its accuracy and applicability.

This is a *PLOS Computational Biology* Methods paper.

## 1 Introduction

Stochastic modelling in systems and synthetic biology has become an indispensable tool to quantitatively understand the intrinsically noisy dynamics within living cells [1]. Intracellular reaction networks typically involve low copy-number species in reactions that fire intermittently at random times, as opposed to continuously. Hence, deterministic models of such networks based on Ordinary Differential Equations (ODEs) fail to capture the essential properties of the system, and stochastic models become necessary [2].

Among the most widely used stochastic models are continuous-time Markov chains (CTMCs) whose states represent the copy-numbers of all species involved in the Chemical Reaction Network (CRN) [3]. If the number of species in the CRN is $n$, the Markov chain evolves over a discrete, possibly infinite, state-space $\mathfrak{X} \subset \mathbb{N}_0^n$ comprising all accessible states. In most applications, the key object of interest is the probability distribution $p(t)$ of the random state $X(t) \in \mathfrak{X}$ at time $t$. This probability distribution evolves in time $t$ according to Kolmogorov's forward equation that is more famously known in the chemical literature as the Chemical Master Equation (CME) (see, e.g., [4], and (8)). The CME is a system of coupled, deterministic ODEs describing the rates of inflow and outflow of probability at each state in the state-space $\mathfrak{X}$. For even very small examples of CRNs, $\mathfrak{X}$ can be very large or infinite, and hence the CME cannot be solved directly despite the linear nature of its constituent ODEs. Hence, one typically estimates CME solutions numerically either by simulating the CTMC trajectories with numerical methods like the Stochastic Simulation Algorithm (SSA) [5] or the modified Next Reaction Method (mNRM) [6], or one models (parts of) the CME asymptotically in various parameter regimes, such as the large copy-number limit, or the large systems limit (see, e.g., [3, 7] and the references therein). Then, Fokker-Planck PDEs govern the evolution of the limiting

densities. Solutions of these PDEs are known to admit DNN approximations which are free from the "Curse of Dimensionality" (CoD), see e.g. [8] and the references there.

The main drawback of simulation-based solvers is that obtaining statistically precise estimates of the CME solution can be very cumbersome, due to the high cost of CTMC trajectory simulation. This led to the development of the *Finite State Projection* (FSP) method [9] that approximately solves the CME by truncating the state-space to a finite, tractable size. The FSP has been successfully used in many important biological studies with stochastic reaction network models. Over time, numerous algorithmic improvements to the original FSP method have been made, using advanced techniques such as Krylov Subspace approximations [10] or Tensor-Train representations [11]. Despite these advances, the scope of FSP's applicability is still fairly limited because of the CoD inherent to the CME for complex CRNs: the dimension of the copy-number space of a large number of species involved in the CRN can be potentially prohibitive. With the algorithmic complexity of deterministic solution methods of the CME scaling exponentially with the number of species $n$, the CoD obviates the efficient numerical treatment of the CME for complex CRNs. In spite of these drawbacks, simulation schemes like the SSA or mNRM combined with FSP and its variants have emerged as the methodology of choice during the past decades for the computational exploration of complex CRNs in systems biology. This is mainly due to the lack of computational schemes that can effectively deal with CoD.

In the past decade, with the ubiquitous emergence of possibly massive, noisy data from natural biological CRNs, and the possibility of *engineering synthetic biological CRNs*, the question of efficient numerical analysis of CRNs has become pivotal. Indeed several tasks in computational biology strongly depend on the availability of *scalable, efficient computational tools to analyse large CRNs*. These include structure and parameter identification in large CRNs, assimilation of observable data into CRN models, Bayesian estimation of non-observable quantities of interest conditional on CRNs, among many others.

Recently, in the context of high-dimensional partial differential equations (PDEs), *deep-learning based numerical approaches* have been found highly effective in dealing with the CoD in these settings and appear efficient in numerical approximation of PDE solutions with high-dimensional state and parameter spaces [12–14]. We refer to the survey [8] and the references therein. Importantly, several types of PDEs considered in these studies also arise from various asymptotic scalings (large copy-numbers, large systems limits) of large CRNs. (e.g. [4, 7, 15, 16]). Furthermore, DNNs have been shown to be at least as expressive as certain tensor-structured formats from numerical multi-linear algebra, which were developed for the CME in [11] (see also [17]).

Motivated by these advances and observations, in this paper we develop and explore corresponding deep-learning approaches for the efficient numerical solution of CMEs and for the related tasks of parameter estimation, and inference.

Before detailing our approach, we remark that leveraging Machine Learning (ML) based approaches for the numerical treatment of complex CRNs is, in our view, natural and critical: CRNs being themselves networks, any viable computational approach should, in some sense, mimic this structure in order to accommodate the complexity of CRNs. This is in line with our previous work on *tensor network based computational methods* (e.g. [11, 18]). On the other hand, ML-based computational methodologies for data assimilation and quantitative prediction of complex systems is currently undergoing intense development. We therefore expect that corresponding advances in computational ML, such as progress in interpretability and training methods for DNNs, will entail corresponding methodological advances in the exploration of large, complex CRNs in biological systems engineering.

Next we briefly describe our ML approach to solving the CME. Instead of estimating the probability $p(t, x)$ for each state $x \in \mathfrak{X}$, one is often interested in learning the expectation of suitable real-valued functions $g$, referred to as the *output* function, under this probability distribution. Therefore, one is interested in the input-output map that associates an initial density $\{p(0, x_0) : x_0 \in \mathfrak{X}\}$ to

$$\mathbb{E}(g(X(t))) = \sum_{x \in \mathfrak{X}} g(x) p(t, x). \tag{1}$$

In the case $\#(\mathfrak{X}) = \infty$ this sum is formal for now. We later will indicate some sufficient conditions for this sum to be well-defined—applying state-space truncation schemes e.g. [9], we may assume that $\#(\mathfrak{X}) < \infty$ holds with a small error in the estimated expectation, which renders the summation finite. For example, if $g(x_1, \ldots, x_n) = x_i^m$, for some $m \in \mathbb{N}_0$ and $i \in \{1, \ldots, n\}$, then the output to be estimated is the $m$-th moment of the random copy-number of the $i$-th species at time $t$, i.e.

$$\mathbb{E}(g(X(t))) = \mathbb{E}(X_i^m(t)).$$

Another relevant example is when $g(x) = \mathbb{1}_A(x)$, the indicator function for some subset $A \subset \mathfrak{X}$ defined as

$$\mathbb{1}_A(x) = \begin{cases} 1 & \text{if } x \in A \\ 0 & \text{otherwise.} \end{cases}$$

Then the output is the probability of the state $X(t)$ being in set $A$ at time $t$

$$\mathbb{E}(g(X(t))) = \mathbb{P}(X(t) \in A).$$

One method of choice to numerically approximate the map $p(0, \cdot) \mapsto \mathbb{E}(g(X(t)))$ is by stochastic simulations generated with the SSA and its variants (e.g. [5, 19–21] and the references therein) combined with ensemble averaging. Generally, this approach mandates a large number of sample paths, to achieve Monte Carlo convergence to reasonable accuracy for $\mathbb{E}(g(X(t)))$ at fixed $t > 0$. In the present paper, we propose DeepCME, a *deep neural network* based methodology to emulate the above-mentioned map. Also in the present approach path simulation is required, during the DNN training phase. However, we find that the number of paths to achieve DNN training generally is lower than by direct use of Monte Carlo estimator combined with stochastic simulations; accuracy is achieved through the generalization properties of DNNs rather than through approximation of admissible sets of initial densities.

As is by now well-known in ML, an essential ingredient in DNN based approaches to emulate high-dimensional maps is the mathematical setup of suitable loss-functions which determine the training process. In DeepCME, we propose a particular loss function which is inspired by other, recent approaches in computational finance (e.g. [12] and the references therein). Specifically, using Kolmogorov's backward equation, Kurtz's random time change formulation [22] and Ito's formula for jump processes, we identify an equation that the output quantity of interest $\mathbb{E}(g(X(t)))$ along with some "policy map" $\mathcal{V}(t, X(t))$ must *uniquely* satisfy for each stochastic trajectory $(X(t))_{t \geq 0}$ almost surely. Minimising a "loss" function that measures the error in this equation, allows us to train a deep neural network to learn the policy map and the quantity of interest $\mathbb{E}(g(X(t)))$ in a reinforcement learning framework. Remarkably, this approach also yields the sensitivities of the quantity of interest $\mathbb{E}(g(X(t)))$ w.r.t. all model parameters. Estimating these parametric sensitivities is important for many applications, but it is considered a difficult problem towards which a lot of research effort has recently been directed [23–31].

This paper is organised as follows. In Section 2 we provide some background on the CTMC model of a reaction network. In Section 3 we present our main results that allow us to cast the problem of solving a CME into the reinforcement learning framework. In Section 4 we describe our deep-learning approach and its implementation in `TensorFlow`. In Section 5 we illustrate this approach with four examples. Finally, in Section 6 we conclude and present directions for future research.

## 2 Preliminaries

### 2.1 The stochastic model

We start by describing the *continuous-time Markov chain* (CTMC) model of a reaction network. Consider a network with $n$ species, denoted by $\mathbf{X}_1, \ldots, \mathbf{X}_n$, that participate in $K$ reactions of the form

$$\sum_{i=1}^{n} v_{ki} \mathbf{X}_i \longrightarrow \sum_{i=1}^{n} v'_{ki} \mathbf{X}_i , \quad k = 1, \ldots, K, \tag{2}$$

where $v_{ki}$ (resp. $v'_{ki}$) is the number of molecules of species $\mathbf{X}_i$ consumed (resp. produced) by reaction $k$. The system's state $x = (x_1, \ldots, x_n) \in \mathbb{N}_0^n$ at any time is the vector of copy-numbers of the $n$ species. As time advances, this state gets displaced by the *stoichiometric* vector $\zeta_k = (v'_{k1} - v_{k1}, \ldots, v'_{kn} - v_{kn})$ when reaction $k$ fires, and this event occurs at rate $\lambda_k(x)$ where $\lambda_k : \mathbb{N}_0^n \to [0, \infty)$ is the propensity function for reaction $k$. Commonly $\lambda_k$ is given by mass-action kinetics [3]

$$\lambda_k(x_1, \ldots, x_n) = c_k \prod_{i=1}^{n} \binom{x_i}{v_{ki}}, \tag{3}$$

with $c_k > 0$ being the associated rate constant.

There are many ways to formally specify the CTMC representing a reaction network. One way is through its generator, which is an operator that captures the rate of change of the probability distribution of the process (see Chapter 4 in [22]). It is given by

$$\mathbb{A}f(x) = \sum_{k=1}^{K} \lambda_k(x)(f(x + \zeta_k) - f(x)), \tag{4}$$

for any $f$ that is a bounded real-valued function on the state-space $\mathfrak{X} \subset \mathbb{N}_0^n$ of the Markov chain. The state-space $\mathfrak{X}$ is assumed to be nonempty and closed under the reaction dynamics, i.e. if $x \in \mathfrak{X}$ and $\lambda_k(x) > 0$ then $(x + \zeta_k)$ is also in $\mathfrak{X}$.

Another way to specify the CTMC is via Kurtz's random time change representation (see Chapter 6 in [22])

$$X(t) = X(0) + \sum_{k=1}^{K} R_k(t)\zeta_k, \tag{5}$$

where $R_k(t)$ is a counting process that counts the number of firings of reaction $k$ in the time-period $[0, t]$. As is customary in trajectory-simulation (e.g. [3, 19]) which will also be required by us for DNN training, we express it in terms of an independent unit rate Poisson process $Y_k$

as e.g. [21]

$$R_k(t) = Y_k \left( \int_0^t \lambda_k(X(s))ds \right). \tag{6}$$

With this representation in place, we consider the CTMC $(X(t))_{t \geq 0}$ on the canonical probability space generated by the independent unit-rate Poisson processes $Y_1, \ldots, Y_K$.

## 2.2 Kolmogorov's forward and backward equations

Let $(X(t))_{t \geq 0}$ be the CTMC representing reaction dynamics with some initial state $X(0) \in \mathfrak{X}$. For any state $x \in \mathfrak{X} \in \mathbb{N}_0^n$, let

$$p(t, x) = \mathbb{P}(X(t) = x) \tag{7}$$

be the probability that the CTMC is in state $x$ at time $t$. These probabilities evolve deterministically in time according to Kolmogorov's forward equation, more widely known as the *Chemical Master Equation* (CME) [3, 4]. The CME is the following system of deterministic linear ODEs

$$\frac{dp(t, x)}{dt} = \sum_{k=1}^K p(t, x - \zeta_k)\lambda_k(x - \zeta_k) - p(t, x)\sum_{k=1}^K \lambda_k(x), \tag{8}$$

for each $x \in \mathfrak{X}$. Note that the number of ODEs in this CME system is equal to $\#(\mathfrak{X})$, the number of elements in $\mathfrak{X}$, which is typically exorbitantly large or even infinite.

Consider an *output* function $g : \mathfrak{X} \to \mathbb{R}$ such that

$$\mathbb{E}(|g(X(T))|) < \infty \tag{9}$$

for some finite time horizon $T > 0$. The Kolmogorov's backward equation [32] describes the evolution of the martingale (w.r.t. the filtration generated by $(X(t))_{t \geq 0}$)

$$V_g(t, x) = \mathbb{E}(g(X(T))|X(t) = x) \tag{10}$$

in the time interval $[0, T]$, and it is given by

$$\frac{\partial}{\partial t} V_g(t, x) = -\mathbb{A}V_g(t, x) = -\sum_{k=1}^K \lambda_k(x)(V_g(t, x + \zeta_k) - V_g(t, x)), \tag{11}$$

with the terminal condition $V_g(T, x) = g(x)$, $x \in \mathfrak{X}$. The backward Eq (10) will play a key role in our development of a deep learning approach for estimating quantities of the form $\mathbb{E}(g(X(T)))$.

In the case where the state-space $\mathfrak{X}$ is finite, i.e. $\#(\mathfrak{X}) = m < \infty$, we can enumerate it as $\mathfrak{X} = \{x^{(1)}, \ldots, x^{(m)}\}$. Then the CTMC generator $\mathbb{A}$ in (11) can be expressed as the $m \times m$ transition rate matrix $Q = [Q_{ij}]$ given by

$$Q_{ij} = \begin{cases} -\sum_{k=1}^K \lambda_k(x^{(i)}) & \text{if } i = j \\ \lambda_k(x^{(i)}) & \text{if } x^{(j)} = x^{(i)} + \zeta_k \text{ for some } k \\ 0 & \text{otherwise.} \end{cases}$$

Here we assume for convenience that all stoichiometry vectors ($\zeta_k$-s) are distinct. Viewing $p(t)$ as the vector $p(t) = (p(t, x^{(1)}), \ldots, p(t, x^{(m)})) \in [0, 1]^{\#(\mathcal{X})}$, we can express the CME (8) as

$$\frac{dp}{dt} = Q^\top p(t), \qquad t \geq 0 \ . \tag{12}$$

Here, $Q_{ij}^\top := Q_{ji}$, $i, j \in 1 : m$. CME (12) admits the closed-form solution

$$p(t) = \exp(tQ^\top)p(0) \quad \text{for any} \quad t \geq 0. \tag{13}$$

Similarly, viewing $V_g(t)$ as the vector $(V_g(t, x^{(1)}), \ldots, V_g(t, x^{(m)})) \in \mathbb{R}^{\#(\mathcal{X})}$, the backward Eq (11) can be solved as

$$V_g(t) = \exp(Q(T - t))g \quad \text{for any} \quad t \in [0, T] \tag{14}$$

where $g$ denotes the vector $g = (g(x^{(1)}), \ldots, g(x^{(m)})) \in \mathbb{R}^{\#\mathcal{X}}$. We are interested in networks where $m = \#(\mathcal{X})$ is extremely large or infinite. Then, numerically computing the matrix exponential in (13) or in (14) is not an option.

## 2.3 Parametric sensitivity analysis

Now consider the situation where the propensity functions depend on a scalar parameter $\theta$ (like reaction rate constant for mass-action kinetics, temperature etc.). Denoting the $\theta$-dependent CTMC as $(X_\theta(t))_{t \geq 0}$, it is often of interest to compute the parametric sensitivity

$$S_\theta(g, T) \quad = \frac{\partial}{\partial \theta} \mathbb{E}(g(X_\theta(T))), \tag{15}$$

of the observed output $\mathbb{E}(g(X_\theta(T)))$ at time $T$. Such sensitivity values are important for many applications and their direct calculation is generally impossible but a number of simulation-based approaches have recently been developed to provide efficient numerical estimation of these sensitivity values; we mention only [23–31].

Theorem 3.3 in [29] proves that

$$S_\theta(g, T) \quad = \sum_{k=1}^K \mathbb{E}\left( \int_0^T \frac{\partial \lambda_k(X_\theta(t), \theta)}{\partial \theta} \left( V_g(t, X_\theta(t) + \zeta_k) - V_g(t, X_\theta(t)) \right) dt \right), \tag{16}$$

where $V_g(t, x)$ is defined by (10) with $X(\cdot)$ replaced by $X_\theta(\cdot)$. The main difficulty in using this formula for computing sensitivities is that the function

$$\Delta_k V_g(t, x) := V_g(t, x + \zeta_k) - V_g(t, x) \tag{17}$$

is unknown and hence it must be estimated "on the fly" by numerically generating auxiliary paths [29]. In the method we develop in this paper we shall "learn" (i.e., emulate by ML techniques) this function using deep neural networks. This would provide a simple direct way to estimate the parameter sensitivity via formula (16). This approach would in fact yield sensitivities w.r.t. *all* the model parameters in one shot, unlike what is afforded by existing sensitivity estimation approaches. In other words once the function $x \mapsto \Delta_k V_g(t, x)$ is available for each $k \in 1 : K$, we can use a common set of simulated trajectories to evaluate Monte Carlo estimators for sensitivities w.r.t. all parameters, based on expression (16), without any extra simulation effort. This is unlike most simulation-based approaches where estimation of each parameter sensitivity requires an additional set of distinct trajectories.

## 3 Main results

In this section we state and prove the key result on which our deep learning approach depends. Recall that our goal is to estimate $\mathbb{E}(g(X(t)))$ (see (1)) which is the same as $V_g(0, x_0)$ (see (10)) if the initial state of the CTMC is $X(0) = x_0$. Also recall the random time-change representation (5) and the definition of the reaction counting process $R_k$ from (6). Henceforth we shall denote the *centred* version of this process as

$$\tilde{R}_k(t) := Y_k\left(\int_0^t \lambda_k(X(s))ds\right) - \int_0^t \lambda_k(X(s))ds\,, \quad k = 1,\ldots,K. \tag{18}$$

This centred process is a local martingale w.r.t. the filtration $\mathcal{F}_X(t)$ generated by $(X(t))_{t\geq 0}$ (see Chapter 1 in [33]).

We now state an assumption that we require for our approach.

**Assumption 3.1 (Non-explosivity of the CTMC)** *Let $(X(t))_{t\geq 0}$ be the CTMC given by (5) with deterministic initial condition $X(0) = x_0$. Let $\mathfrak{X} \subset \mathbb{N}_0^n$ denote the state-space of this CTMC and let $\mathcal{F}_X(t)$ be the filtration it generates. If $\tau_M$ is the $\mathcal{F}_X(t)$-stopping time defined by*

$$\tau_M = \inf\{t \geq 0 :\| X(t) \|\geq M\},$$

*then $\tau_M \to \infty$ almost surely as $M \to \infty$.*

**Remark 3.2** *There are a number of works in the literature that provide sufficient conditions for this non-explosivity condition to hold, subject to the form of the propensity functions (see for example in [34–36] and the references therein). Under the no-explosion assumption a probability distribution p(t) satisfying the CME exists uniquely (see e.g. Lemma 1.23 in [33]).*

Next we present the main result on which our deep learning approach is based.

**Theorem 3.3 (Expected output and policy map characterization)** *Suppose Assumption 3.1 holds for the CTMC $(X(t))_{t\geq 0}$ and the output function $g : \mathfrak{X} \to \mathbb{R}$ satisfies (9). Let $\mathcal{Y}$ be a real number and let $\mathcal{V}(t,x) = (\mathcal{V}_1(t,x),\ldots,\mathcal{V}_K(t,x))$ be a measurable $\mathbb{R}^K$-valued function on $[0,T] \times \mathfrak{X}$ such that the following relation holds almost surely*

$$g(X(T)) = \mathcal{Y} + \sum_{k=1}^K \int_0^T \mathcal{V}_k(t, X(t))d\tilde{R}_k(t). \tag{19}$$

*Then $\mathcal{Y}$ and $\mathcal{V}(t,x)$ exist uniquely and they can be identified as*

$$\mathcal{Y} = \mathbb{E}(g(X(T))) \quad \text{and} \quad \mathcal{V}(t,x) = (\Delta_1 V_g(t,x),\cdots,\Delta_K V_g(t,x)) \tag{20}$$

*where $V_g(t,x)$ is given by (10) and the difference operator $\Delta_k$ is as in (17).*

**Remark 3.4 (Connection to our deep learning approach)** *Before we prove this theorem, we briefly describe how this result translates into our deep learning approach, details of which will be provided in Section 4. We can view $x \mapsto \mathcal{V}(t,x)$ as the "policy map" (in the parlance of reinforcement learning) that decides actions based on the current time-state pair (t, x), and depending on these actions the constant initial value $\mathcal{Y}$ is evolved in the time-interval $[0,T]$ according to the r. h.s. of (19) for any CTMC trajectory $(X(t))_{t\geq 0}$. Theorem 3.3 shows that the only way the final outcome of this evolution matches g(X(T)) at time T, is when $\mathcal{Y}$ is exactly the expected output $\mathbb{E}(g(X(T)))$, and the policy map $\mathcal{V}(t,x)$ is exactly $(\Delta_1 V_g(t,x),\ldots,\Delta_K V_g(t,x))$ where $\Delta_k V_g(t,x)$ is the difference in the expected output $\mathbb{E}(g(X(T)))$ at time T, due to a single firing of reaction k at time t with system's state at x = X(t).*

*Using the modified next reaction method [6], one can easily generate trajectories of the CTMC $(X(t))_{t\geq 0}$ along with the associated centred reaction counting processes $(\tilde{R}_1(t),\ldots,\tilde{R}_K(t))_{t\geq 0}$. For each such trajectory, relation (19) can be interpreted in terms of*

*known and unknown quantities as*

$$\underbrace{g(X(T))}_{\text{known}} = \underbrace{\mathcal{Y}}_{\text{unknown}} + \sum_{k=1}^{K} \int_0^T \underbrace{\mathcal{V}_k(t, X(t))}_{\text{unknown}} \underbrace{d\tilde{R}_k(t)}_{\text{known}}. \tag{21}$$

*We represent the unknown map $(t, x) \mapsto \mathcal{V}(t, x)$ by a DNN and consider unknown $\mathcal{Y}$ as an optimisation variable. Then by minimising a "loss" function $\mathcal{L}(\mathcal{Y}, \mathcal{V})$ that measures the discrepancy in relation* (21) *we try to recover the optimal values of $\mathcal{Y}$ and $\mathcal{V}$ that are given by* (20). *This allows us to estimate the output of interest $\mathbb{E}(g(X(T)))$ (as $\mathcal{Y}$) and also its parametric sensitivities by substituting $\mathcal{V}_k(t, x)$ for $\Delta_k V_g(t, x)$ in* (16).

*Observe that in traditional simulation-based estimation approaches, each simulated trajectory contributes with a small weight (viz. reciprocal of the sample size) to the Monte Carlo estimator for the output or one of its parameter sensitivities. This is quite unlike the proposed deep learning approach where each trajectory specifies an almost sure relationship between the unknown quantities that determine both the output and all its parameter sensitivities. Hence the deep learning approach is able to extract more information out of a small number of simulated trajectories as our examples in Section 5 illustrate.*

**Proof**.[Proof of Theorem 3.3] We prove this result in two steps. We first show that $\mathcal{Y}$ and $\mathcal{V}(t, x)$ given by (20) satisfy (19) almost surely. Then, we prove that if another such pair $(\hat{\mathcal{Y}}, \hat{\mathcal{V}}(t, x))$ satisfying (19) exists then we must necessarily have $\hat{\mathcal{Y}} = \mathcal{Y}$ and $\hat{\mathcal{V}}(t, x) = \mathcal{V}(t, x)$.

Applying Ito's formula for jump Markov processes to $V_g(t, X(t))$ we obtain

$$V_g(T, X(T)) = V_g(0, x_0) + \int_0^T \frac{\partial}{\partial t} V_g(t, X(t)) dt + \sum_{k=1}^{K} \int_0^T \Delta_k V_g(t, X(t)) dR_k(t).$$

Using Kolmogorov's backward Eq (11) and simplifying we get

$$V_g(T, X(T)) = V_g(0, X(0)) + \sum_{k=1}^{K} \int_0^T \Delta_k V_g(t, X(t)) d\tilde{R}_k(t). \tag{22}$$

Noting that $V_g(T, X(T)) = g(X(T))$ and $V_g(0, X(0)) = \mathbb{E}(g(X(T)))$ we see that (19) holds with $\mathcal{Y}$ and $\mathcal{V}(t, x)$ given by (20).

Now let $(\hat{\mathcal{Y}}, \hat{\mathcal{V}}(t, x))$ be another pair satisfying (19), i.e.

$$g(X(T)) = \hat{\mathcal{Y}} + \sum_{k=1}^{K} \int_0^T \hat{\mathcal{V}}_k(t, X(t)) d\tilde{R}_k(t).$$

We subtract (22) from this equation to obtain

$$\Delta\hat{\mathcal{Y}} + \sum_{k=1}^{K} \int_0^T \Delta\hat{\mathcal{V}}_k(t, X(t)) d\tilde{R}_k(t) = 0 \tag{23}$$

where

$$\Delta\hat{\mathcal{Y}} = \hat{\mathcal{Y}} - \mathbb{E}(g(X(T))) \quad \text{and} \quad \Delta\hat{\mathcal{V}}_k(t, X(t)) = \hat{\mathcal{V}}_k(t, X(t)) - \Delta_k V_g(t, X(t)).$$

Note that

$$m(t) := \Delta\hat{\mathcal{Y}} + \sum_{k=1}^{K} \int_0^t \Delta\hat{\mathcal{V}}_k(s, X(s)) d\tilde{R}_k(s)$$

is a local martingale w.r.t. the filtration $\mathcal{F}_X(t)$ generated by $(X(t))_{t \geq 0}$ as it is defined as a sum of stochastic integrals whose integrands are adapted to $\mathcal{F}_X(t)$ and whose integrators are local martingales w.r.t. $\mathcal{F}_X(t)$ (see Appendix A.3 in [33]).

If $\tau_M$ is the stopping time defined in Assumption 3.1, then the stopped process $m(t \wedge \tau_M)$ is a martingale, where $a \wedge b := \min\{a, b\}$. Applying Doob's maximal inequality [22] on the sub-martingale $|m(t \wedge \tau_M)|$ we obtain

$$\mathbb{E}\left[\left(\sup_{0 \leq t \leq T \wedge \tau_M} |m(t)|\right)^2\right] \leq 4\mathbb{E}(m(T \wedge \tau_M)^2). \tag{24}$$

Note that terms on both sides of the inequality are monotonically increasing in $M$. This monotonicity is obvious for the term on the l.h.s. and for the term on the r.h.s. it follows from the conditional Jensen's inequality and from the martingale property

$$\begin{aligned}
\mathbb{E}(m(T \wedge \tau_{M+1})^2) &= \mathbb{E}[\mathbb{E}(m(T \wedge \tau_{M+1})^2 | \mathcal{F}_X(T \wedge \tau_M))] \\
&\geq \mathbb{E}[(\mathbb{E}(m(T \wedge \tau_{M+1}) | \mathcal{F}_X(T \wedge \tau_M)))^2] \\
&= \mathbb{E}(m(T \wedge \tau_M)^2).
\end{aligned}$$

Letting $M \to \infty$ and using the monotone convergence theorem on both sides of (24) we obtain

$$\mathbb{E}\left[\left(\sup_{0 \leq t \leq T} |m(t)|\right)^2\right] \leq 4\mathbb{E}(m(T)^2)$$

where we have used the fact that $\tau_M \to \infty$ as $M \to \infty$ due to Assumption 3.1. Relation (23) informs us that $m(T) = 0$ almost surely and hence

$$\mathbb{E}\left[\left(\sup_{0 \leq t \leq T} |m(t)|\right)^2\right] = 0.$$

This is sufficient to conclude that $\Delta\hat{\mathcal{Y}} = 0$ and $\Delta\hat{\mathcal{V}}_k(t, X(t)) = 0$ for any $t \in [0, T]$.

As this holds for any CTMC trajectory $(X(t))_{t \geq 0}$, we must have $\Delta\hat{\mathcal{V}}_k(t, x) = 0$ for any $(t, x) \in [0, T] \times \mathfrak{X}$. This completes the proof of this theorem.

## 4 DeepCME: Deep learning formulation for CME

In this section we detail our deep learning method for solving CME, referred to as *DeepCME*. We have computationally implemented this method using the machine learning library `TensorFlow` [37]. Our source code for generating the ensuing numerical experiments is available at GitHub: https://github.com/ankitgupta83/DeepCME.

As outlined in Remark 3.4, our approach is based on the almost sure relationship established in Theorem 3.3. Even though this result was presented for a single output function $g(x)$, it can be easily extended for a vector-valued function $g(x) = (g_1(x), \ldots, g_R(x))$ by considering the unknown variable $\mathcal{Y}$ as a $R$-dimensional vector and the unknown map $\mathcal{V}(t, x)$ that takes a time-state pair $(t, x)$ as input and produces an output in the space of $R \times K$ matrices. Such an extension is useful because in most applications one is interested in estimating multiple statistical properties (like means, variances, covariances etc.) of the CME solution $p(T, \cdot)$.

We now define the "loss" function $\mathcal{L}(\mathcal{Y}, \mathcal{V})$ that measures the discrepancies in the $R$ almost sure relations given by (21). Let $\boldsymbol{L} : \mathbb{R}^R \to [0, \infty)$ be the following continuously differentiable

function

$$L(x_1, \ldots, x_R) = \sum_{i=1}^{R} \phi\left(\frac{x_i}{\Delta_i}\right)$$

where $\Delta = (\Delta_1, \ldots, \Delta_R)$ is a vector of positive threshold values and

$$\phi(x) = \begin{cases} x^2 & \text{if } |x| < 1 \\ 2|x| - 1 & \text{otherwise.} \end{cases}$$

We define the loss function as

$$\mathcal{L}(\mathcal{Y}, \mathcal{V}) = \mathbb{E}\left[L\left(g(X(T)) - \mathcal{Y} - \sum_{k=1}^{K} \int_0^T \mathcal{V}_k(t, X(t)) d\tilde{R}_k(t)\right)\right], \tag{25}$$

where the expectation is estimated by computing the sample mean over a finite *batch* of "training" trajectories. During the training process this loss function is minimised in order to learn the optimal $\mathcal{Y}$, which estimates our expectations of interest

$$\mathbb{E}(g(X(T))) = (\mathbb{E}(g_1(X(T))), \cdots, \mathbb{E}(g_R(X(T)))),$$

and the optimal matrix-valued policy map $\mathcal{V}(t, x)$ (see Remark 3.4). This policy map will enable us to estimate sensitivities of the quantities of interest w.r.t. all the model parameters as discussed in Section 2.3. The threshold values $\Delta = (\Delta_1, \ldots, \Delta_R)$ help in neutralising the disparities in the relative magnitudes of the estimated quantities and the discrepancies in the corresponding almost sure relations. The loss function minimisation is performed with the *stochastic gradient descent* (SGD) algorithm that makes use of the automatic differentiation routines that are built in `TensorFlow`. Differentiability properties of the function $L$ which defines the loss function are important for convergence of the SGD iterations. Our choice of $\phi(x)$ makes $L(x_1, \ldots, x_R)$ a differentiable combination of $\mathcal{L}_1$ norm (for components with absolute values greater than 1) and $\mathcal{L}_2$ norm squared (for components with absolute values strictly less than 1). Having such a combination makes the training more robust and promotes sparsity.

In DeepCME we first encode the matrix-valued policy map $(t, x) \mapsto \mathcal{V}(t, x)$ by a DNN and we include $\mathcal{Y}$ as a vector of trainable variables. Then a batch of training trajectories is generated, and based on $\mathcal{Y}$ and the DNN-encoded policy map $\mathcal{V}(t, x)$, the loss function is evaluated for this training data by measuring the discrepancy (according to (25)) in the almost sure relationship presented in Theorem 3.3. Keeping the training data fixed, this loss function is then minimised by adjusting $\mathcal{Y}$ and the DNN with SGD for a given number of iterations. Once these iterations are complete, $\mathcal{Y}$ provides estimates for the expectations of interest and their parametric sensitivities can be estimated by evaluating Monte Carlo estimators based on expression (16), using the DNN-encoded policy map and the training trajectories.

In the next two subsections we elaborate more on the DNN encoding of the policy map and the loss function computation based on simulated trajectories.

## 4.1 DNN encoding of the policy map

Recall from Section 2.2 that if the state-space is finite then $V_g(t, x)$ can in principle be found by exponentiating the transition rate matrix $Q$ multiplied with $(T - t)$ (see (14)). Hence, if $\lambda = \lambda_1 + i\lambda_2$ is an eigenvalue of $Q$, then on the associated eigenspace we would expect that the dependence of $V_g(t, x)$ on time $t$ is given by $e^{\lambda(T-t)} = e^{\lambda_1(T-t)}(\cos(\lambda_2(T-t)) + i\sin(\lambda_2(T-t)))$.

Motivated by this rationale, rather than passing the time-values $t$ directly as inputs to the DNN that encodes $\mathcal{V}(t, x)$, we shall pass *temporal features* of the form

$$\mathcal{T}(t) = (e^{\lambda_{11}(T-t)}, \ldots, e^{\lambda_{r1}(T-t)}, \sin(\lambda_{12}(T-t) + \psi_1), \ldots, \sin(\lambda_{r2}(T-t) + \psi_r)), \qquad (26)$$

where $\lambda_{11}, \ldots, \lambda_{r1}, \lambda_{12}, \ldots, \lambda_{r2}$ are $2r$ trainable variables that represent the $r$ dominant eigenvalues of the generator of the CTMC. Additionally, $r$ trainable variables $\psi_1, \ldots, \psi_r$ are included to represent 'phase shifts'. Problem-specific temporal features, like the ones we consider, have been successfully employed in existing deep learning methods for ODE-based reaction network models (see, e.g., [38] and the references therein). Note that the mapping between time $t$ and the temporal features $\mathcal{T}(t)$ is one-to-one and hence no information is lost by substituting time inputs with temporal features.

We encode the policy map $(t, x) \mapsto \mathcal{V}(t, x)$ as a fully connected *feedforward* deep neural network whose architecture is schematically shown in Fig 1. This neural network consists of an input layer, $L$ hidden layers and an output layer. Mathematically, DNNs $\Phi$ considered here are determined by a tuple

$$\Phi = ((\hat{T}_1, \rho_1), \ldots, (\hat{T}_{L+1}, \rho_{L+1})), \qquad (27)$$

where in layer $\ell = 1, \ldots, L+1$, the map $\hat{T}_\ell : \mathbb{R}^{N_{\ell-1}} \to \mathbb{R}^{N_\ell}$ is an affine transformation i.e. $\hat{T}_\ell(x) = W_\ell x + b_\ell$, with *weight matrix* $W_\ell \in \mathbb{R}^{N_\ell \times N_{\ell-1}}$, and *bias vector* $b_\ell \in \mathbb{R}^{N_\ell}$. As mentioned, in the presently considered DNNs, the input layer takes the temporal features $\mathcal{T}(t)$ and the state vector $x = (x_1, \ldots, x_n)$.

The nonlinearities $\rho_\ell : \mathbb{R}^{N_\ell} \to \mathbb{R}^{N_\ell}$ in (27) act on vectors in $\mathbb{R}^{N_\ell}$ component-wise, with possibly different activations at each layer. The number $L + 1$ denotes the *number of layers* (sometimes referred to as depth) of the DNN $\Phi$, and $L$ denotes the *number of hidden layers* of DNN $\Phi$.

With the DNN $\Phi$, we associate a *realization*, i.e., a map

$$R(\Phi) : \mathbb{R}^{\mathbb{N}_0} \to \mathbb{R}^{\mathbb{N}_{L+1}}, \quad \text{where} \quad R(\Phi) := \rho_{L+1} \circ \hat{T}_{L+1} \circ \ldots \circ \rho_1 \circ \hat{T}_1.$$

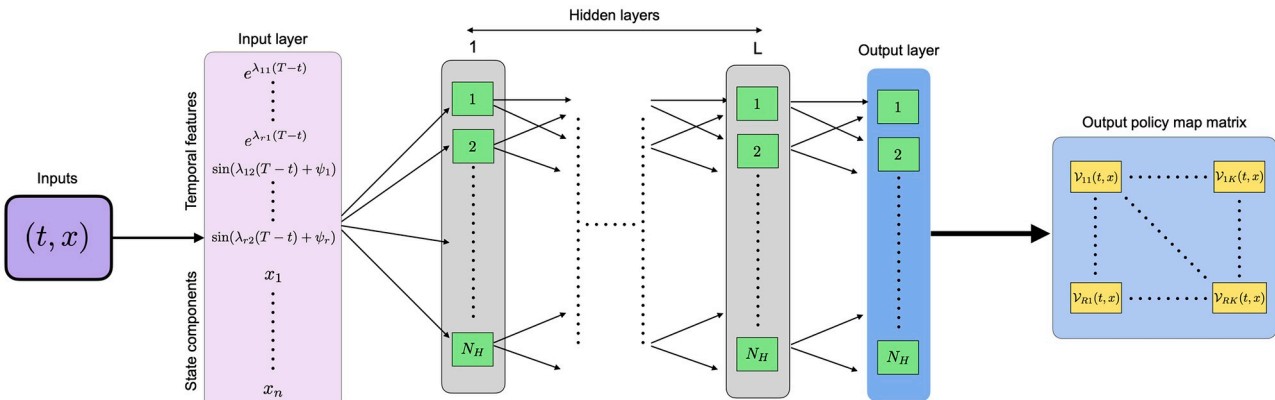

**Fig 1. Architecture of the neural network.** DNN architecture to encode the matrix-valued map $(t, x) \mapsto \mathcal{V}(t, x)$. The inputs $(t, x)$ are passed to an input layer, which leaves the state values $x$ unchanged but activates a dictionary of *temporal features* (26). The resulting output is propagated through a DNN with $L$ fully connected hidden layers, and an additional output layer with each layer having $N_H$ nodes. For simplicity, we assume no sparsity in the weight matrices and the bias vectors of these layers. In the final step, the output from the output layer is cast into the policy-map matrix $\mathcal{V}(t, x)$ corresponding to inputs $(t, x)$. In Section 4.2 we describe how the loss function can be computed using this matrix-valued map for a batch of stochastic trajectories.

The relation between the DNN parameters $\Phi$ and its realisation $R(\Phi) : \mathbb{R}^{N_0} \to \mathbb{R}^{N_{L+1}}$ as a map is not one-to-one: for several choices of $\Phi$, realizations $R(\Phi)$ may give rise to the same map $R(\Phi) : \mathbb{R}^{N_0} \to \mathbb{R}^{N_{L+1}}$. This *over-parametrization* of DNNs is well-known to cause multiple minima in loss functions of DNN parameters, and to obstruct use of efficient optimisation algorithms in numerical DNN training.

The goal of DNN approximations is to provide a parsimonious surrogate map $R(\Phi)$ for many-parametric, input-output maps which are not explicitly known and are accessible computationally only through possibly noisy evaluations.

The input layer transforms the time-value $t$ into temporal features (26) but leaves the state vector $x = (x_1, \ldots, x_n)$ unchanged. For the layers, we assume fixed width, i.e., that each layer consists of $N_H$ nodes (including the output layer). We also assume that no activation is applied at the output layer, i.e. $\rho_{L+1}$ is the identity function, and all activations in the hidden layers are equal, i.e. for $\ell = 1, \ldots, L$ and for $i = 1, \ldots, N_H$, $\varrho = (\rho_\ell)_i : \mathbb{R} \to \mathbb{R}$. In the ensuing numerical examples, we employ the so-called ReLU-*activation* for the hidden layers, which is given by

$$\varrho(x) := \mathrm{ReLU}(x) = \max\{x, 0\}, \quad x \in \mathbb{R} . \tag{28}$$

**Remark 4.1** *More generally, for $k \in \mathbb{N}$, we may choose the activations $\varrho^k(x)$, observing that increasing the value of $k$ increases differentiability of realizations of the DNN $\Phi$. This may be of relevance in cases where the diffusion limits for large copy number counts of particular species imply higher smoothness of the map $x \mapsto p(T, x)$.*

## 4.2 Loss function computation based on the training data

To numerically evaluate the loss function, we require simulated training trajectories of the form $(X(t), \tilde{R}(t))_{t \geq 0}$ where $X$ denotes the CTMC and each $\tilde{R} = (\tilde{R}_1, \ldots, \tilde{R}_K)$ is the vector of centred reaction counting processes defined by (18). Such trajectories can be easily generated with Anderson's modified next reaction (mNRM) method [6]. We discretise the time-interval $[0, T]$ as

$$0 = t_0 < t_1 < \cdots < t_J = T .$$

Based on this partition, each simulated trajectory can be viewed as a collection of $J + 1$ *triplets*

$$(t_j, X(t_j), \tilde{R}(t_j)), \quad j = 0, \ldots, J.$$

For each $j$ we pass the time-state pair $(t_j, X(t_j))$ as input to the DNN-encoded matrix valued policy map to obtain $\mathcal{V}(t_j, X(t_j))$. This allows us to compute $\mathcal{Y}_j$ iteratively as

$$\mathcal{Y}_j = \mathcal{Y}_{j-1} + \mathcal{V}(t_{j-1}, X(t_{j-1}))(\tilde{R}(t_j) - \tilde{R}(t_{j-1})) \quad \text{for} \quad j = 1, \ldots, J,$$

with $\mathcal{Y}_0 = \mathcal{Y}$. Here each $\mathcal{Y}_j$ is a $R \times 1$ vector, $\mathcal{V}(t_{j-1}, X(t_{j-1}))$ is a $R \times K$ matrix and $(\tilde{R}(t_j) - \tilde{R}(t_{j-1}))$ is a $K \times 1$ vector. Following this scheme we can compute $\mathcal{Y}_j^{(q)}$ for the $q$-th simulated trajectory $(X^{(q)}(t), \tilde{R}^{(q)}(t))_{t \geq 0}$. With $M$ such i.i.d. trajectories, the *loss function* (25) can be estimated as

$$\hat{\mathcal{L}}(\mathcal{Y}, \mathcal{V}) := \frac{1}{M} \sum_{q=1}^{M} L\big(g(X^{(q)}(T)) - \mathcal{Y}_J^{(q)}\big). \tag{29}$$

Here, we made use of (23).

**Remark 4.2** *In the loss function* (25) *and its MC estimate* (29)*, one could add a* sparsity-promoting regularization term, *in which case* (25) *would become*

$$\hat{\mathcal{L}}\left(\mathcal{Y}, \mathcal{V}\right) \coloneqq \frac{1}{M}\sum_{q=1}^{M}\boldsymbol{L}\big(g(X^{(q)}(T)) - \mathcal{Y}_J^{(q)}\big) + \mu\mathcal{P}(\Phi) \;. \tag{30}$$

*Here,* $\mu \geq 0$ *is a penalty parameter and* $\mathcal{P}(\Phi)$ *promotes sparsity in weights* $W_\ell$ *and biases* $b_\ell$ *comprising* $\Phi$. *In the numerical experiments we report we did not use this device.*

**Remark 4.3** *When the time-interval* $[0, T]$ *is large, instead of using a single DNN to approximate the policy map* $(t, x) \mapsto \mathcal{V}(t, x)$*, it may beneficial to employ multiple temporal DNNs that are uniformly distributed in the time-interval* $[0, T]$*. All these DNNs have the same structure, as shown in* Fig 1. *If* $N_T$ *such DNNs are employed, then we use the m-th DNN to represent the policy map* $(t, x) \mapsto \mathcal{V}(t, x)$ *for* $t \in [(m-1)\delta, m\delta)$ *where* $m = 1, \ldots, N_T$ *and* $\delta = T/N_T$*. Distributing DNNs across time would reduce the complexity of the policy map (as a function of time t) that is needed to be learned. This is helpful in scenarios where the stochastic dynamics has intricate temporal features, such as oscillations.*

## 5 Examples

We now present four examples to illustrate our *DeepCME* method. All these examples are reaction networks with $n$ species, denoted by $\mathbf{X}_1, \ldots, \mathbf{X}_n$, and $2n$ reactions. By varying $n$, we shall investigate how the DeepCME method performs as the network gets larger and compare its performance with simulation based methods.

In all the examples, we assume that all the species have zero copy-numbers initially, and we consider two output functions $g_1(x) = x_n$ and $g_2(x) = x_n^2$ whose expectation is to be estimated under the probability distribution given by the CME solution at time $T = 1$. In other words, we shall use DeepCME to estimate the first two moments of the copy-number of species $\mathbf{X}_n$ at time $T$, viz.

$$\mathbb{E}(g_1(X(T))) = \mathbb{E}(X_n(T)) \quad \text{and} \quad \mathbb{E}(g_2(X(T))) = \mathbb{E}(X_n^2(T)). \tag{31}$$

We shall compare these estimates to those obtained by simulating 1000 CTMC trajectories with mNRM [6]. Our method DeepCME also yields estimates of the sensitivities of the estimated moments (31) w.r.t. all model parameters. We plot these estimates and compare them with the estimates obtained via the simulation-based *Bernoulli Path Algorithm* (BPA) [29]. These latter estimates are based on a sample of size 1000 and for each sample BPA requires generation of a certain number of auxiliary paths (see Section 2.3) which we set to be 10 in our examples.

In all the examples, we encode the policy map $(t, x) \mapsto \mathcal{V}(t, x)$ as a DNN with $L = 2$ hidden layers and $N_H = 4$ nodes per layer (see Fig 1), irrespective of the number of species $n$. The activation function for all hidden layer nodes is ReLU$(x)$ (see (28)) and we choose $r = 1$ for the temporal features (26) to transform the time-values. For loss function computation, we partition the time-interval $[0, T]$ into $J = 50$ equal size time-increments.

The neural network is trained with a training batch of $M = 100$ trajectories generated a priori with mNRM (see Section 4), and another such batch of $M = 100$ trajectories is used for validation. We display the loss function for the validation trajectories to track the training process. To facilitate comparison across network sizes, we normalise all the loss function trajectories to be one at the start of training. Note that the definition of our loss function (25) depends on certain threshold values $\Delta = (\Delta_1, \Delta_2)$. We choose these values as

$$\Delta_j = 1 + |\hat{\mu}_j| + 2\hat{\sigma}_j$$

where $\hat{\mu}_j$ (resp. $\hat{\sigma}_j$) denotes the sample mean (resp. standard deviation) of the values of the output function $g_j$ for the trajectories in the training batch. Finally, to gauge the computational efficiency of DeepCME we compare the total *central processing unit (CPU)* time it requires (including the time to generate training and validation trajectories) to the total CPU time required by simulation-based approaches (mNRM and BPA) to estimate the expectations (31) and all its parameter sensitivities. All the computations were performed on the Euler computing cluster of ETH Zurich [39].

## 5.1 Independent birth death network

As our first example (see Fig 2A), we consider a network of $n$ species that are all undergoing independent birth-death reactions

$$\emptyset \xrightarrow{\;k\;} \mathbf{X}_j \xrightarrow{\;\gamma\;} \emptyset \quad \text{for} \quad j = 1, \dots, n.$$

We set $k = 10$ and $\gamma = 1$. The propensity functions obey mass-action kinetics and are hence affine functions of the state variable $x$.

For $n = 5, 10, 20$ species, we apply DeepCME to this reaction network by training the neural network for $10'000$ SGD iterations. Based on the trained neural network, we compute estimates of the first two moments (31) and their sensitivities to both the model parameters $k$ and $\gamma$. We also estimate these quantities with simulation-based methods (mNRM and BPA) with 1000 samples, and since the propensity functions are linear we can compute these quantities exactly as well. In plots shown in Fig 2D, 2E and 2F, we compare the estimates from all these approaches for various values of $n$. Observe that DeepCME is in general quite accurate in

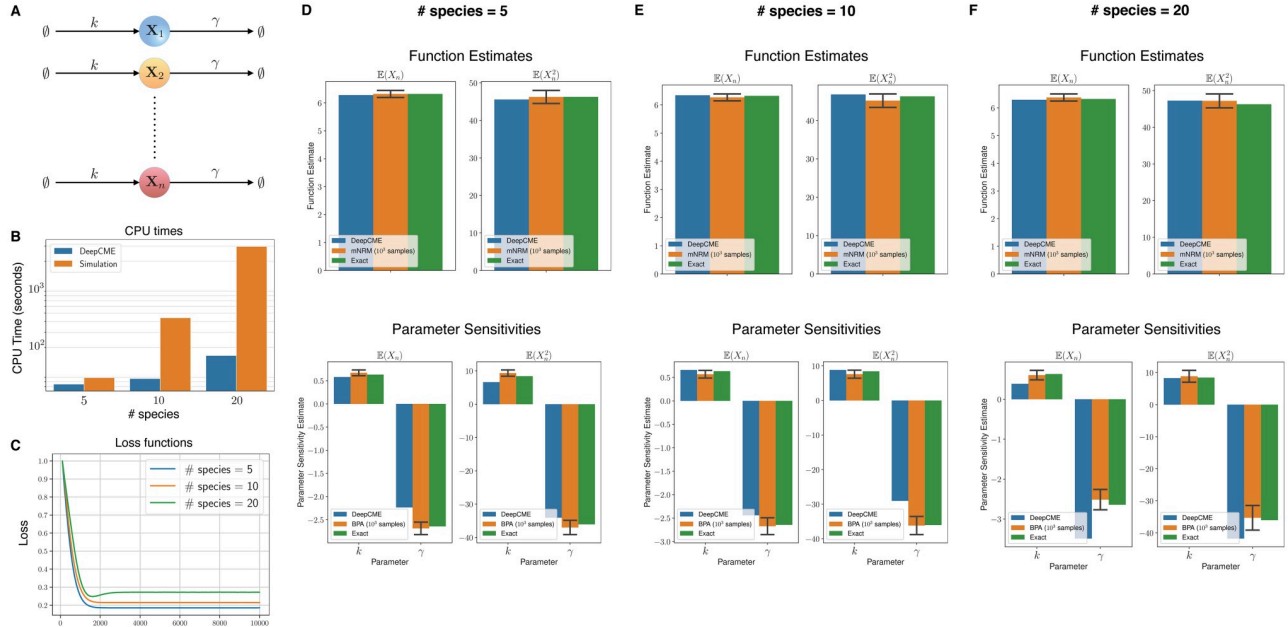

**Fig 2. Independent birth death network.** (A) Depicts the network with $n$ species and $2n$ reactions with mass-action kinetics. (B) The CPU times are shown for DeepCME for different values of $n$ (denoted as # species), and for comparison the time needed by simulation based methods (mNRM for function estimates and BPA for parameter sensitivities) with 1000 trajectories is also indicated. (C) Plots the validation loss function w.r.t. training steps for various $n$ values. Panels (D-F) show estimates for the function values ($\mathbb{E}(X_n(T))$ and $\mathbb{E}(X_n^2(T))$) at T = 1 and the parameter sensitivities. The estimates with simulation based methods are shown as 95% confidence intervals with 1000 samples.

estimating both the moments and their parametric sensitivities, but there are a few cases where the error is significant (e.g. sensitivity w.r.t. $\gamma$ for $\mathbb{E}(X_n^2(T))$ and $n = 20$). These errors can in principle be reduced by employing a different neural network to encode the policy map. In our experience, these errors were also reduced in some cases by including a sparsity promoting term in the loss function (see Remark 4.2) but the result was highly sensitive to the relative weight (i.e. parameter $\mu$ in (30)) of this term.

The CPU time required by DeepCME and simulation-based methods for obtaining moment and sensitivity estimates are plotted in Fig 2B. Note that the CPU time for simulation-based methods grows linearly with the network size $n$, but for DeepCME this growth is sub-linear owing to the fixed structure of the underlying neural network. Despite this fixed structure, the validation loss function trajectories for DeepCME are similar for all $n$ (see Fig 2C), indicating that the training process has low dependence on the number of species, probably because the species are evolving independently.

## 5.2 Linear signalling cascade

Our second example is a linear cascade with $n$-species (see Fig 3A), where species $\mathbf{X}_i$ catalyses the production of species $\mathbf{X}_{i+1}$. The $2n$ reactions are given by

$$\emptyset \xrightarrow{\beta_0} \mathbf{X}_1, \qquad \mathbf{X}_i \xrightarrow{k} \mathbf{X}_{i+1} \quad \text{for} \quad i = 1, \ldots, n-1 \quad \text{and} \quad \mathbf{X}_j \xrightarrow{\gamma} \emptyset \quad \text{for} \quad j = 1, \ldots, n.$$

We set $\beta_0 = 10$, $k = 5$ and $\gamma = 1$. As in the previous example, all the propensity functions obey mass-action kinetics and are hence affine functions of the state.

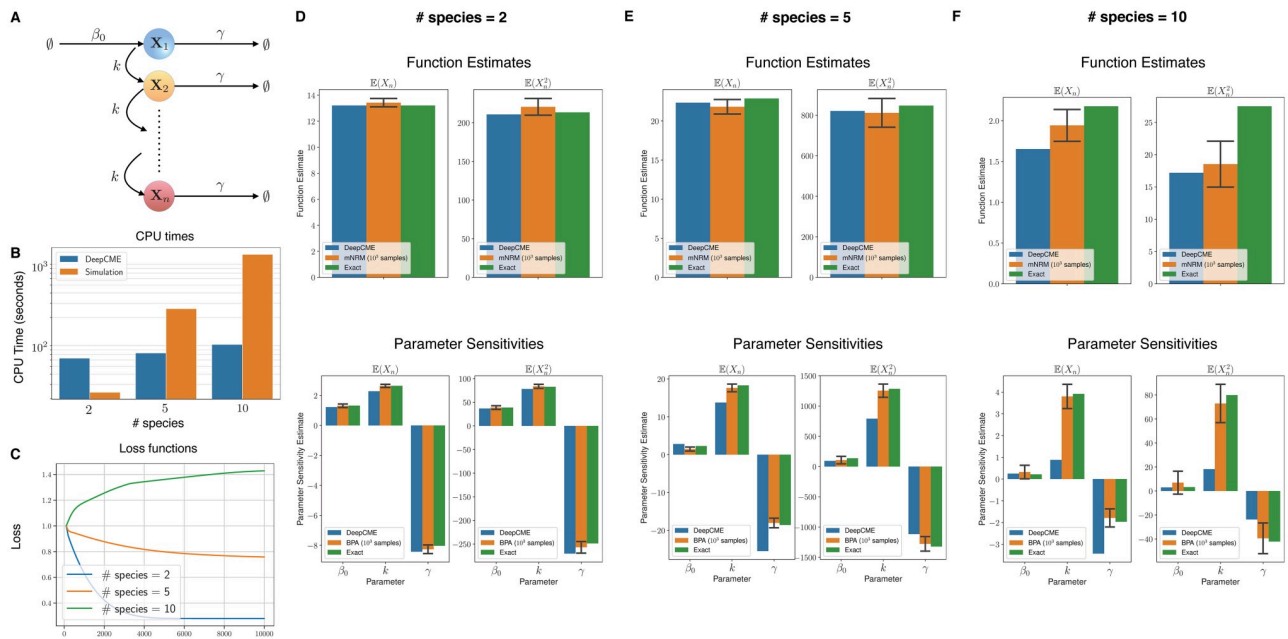

**Fig 3. Linear signalling cascade.** (A) Depicts the network with $n$ species and $2n$ reactions with mass-action kinetics. (B) The CPU times are shown for DeepCME for different values of $n$ (denoted as # species), and for comparison the time needed by simulation based methods (mNRM for function estimates and BPA for parameter sensitivities) with 1000 trajectories is also indicated. (C) Plots the validation loss function w.r.t. training steps for various $n$ values. Panels (D-F) show estimates for the function values ($\mathbb{E}(X_n(T))$ and $\mathbb{E}(X_n^2(T))$) at T = 1 and the parameter sensitivities. The estimates with simulation based methods are shown as 95% confidence intervals with 1000 samples.

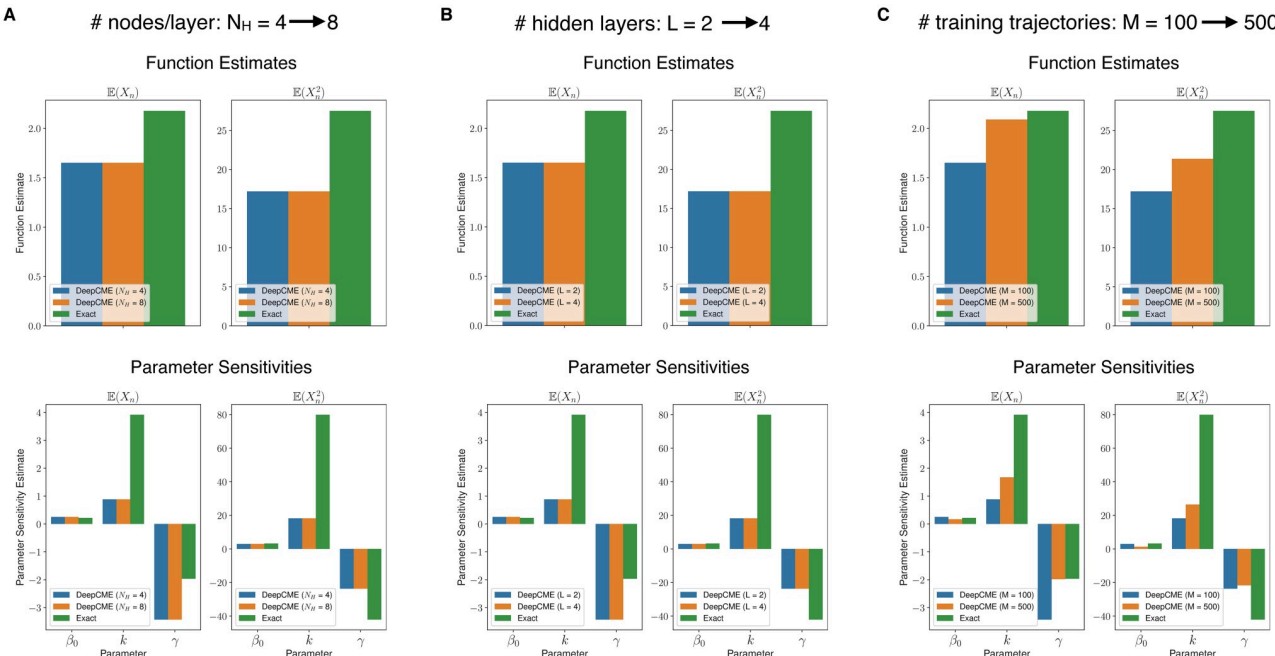

**Fig 4. Linear signalling cascade (continued).** In panels (A) and (B) we illustrate that the accuracy of the DeepCME estimates *remains unaltered* when the DNN shape parameters—$N_H$ (number of nodes per layer) and $L$ (number of hidden layers)—are doubled from their default values of $N_H = 4$ and $L = 2$. In panel (C) we illustrate that the accuracy of these estimates *improves* when the number of training trajectories is increased from $M = 100$ to $M = 500$.

For number of species $n = 2, 5, 10$, we apply DeepCME to this reaction network by training the neural network for 10′000 SGD iterations. Then we compute the moment estimates (31) and their sensitivities to all the model parameters. These quantities are also estimated with simulation-based methods (mNRM and BPA) with 1000 samples, and as with the previous example, the linearity of the propensity functions enables us to compute these quantities exactly as well. In the plots shown in Fig 3D, 3E and 3F, we compare the estimates from all these approaches for various values of $n$. Observe that DeepCME is accurate in estimating the moments but some of the parameter sensitivity estimates are not very accurate (e.g. sensitivity w.r.t. $k$ for $\mathbb{E}(X_n^2(T))$ and $n = 10$). This is because the training process is not successful, as indicated by the validation loss function trajectories shown in Fig 3C. The CPU times for DeepCME and simulation-based methods are plotted in Fig 3B, and as expected they show sub-linear growth w.r.t. $n$ for the former but linear growth for the latter.

It is natural to ask if the accuracy of the estimates provided by DeepCME for $n = 10$ can be improved by making the DNN "deeper" (by increasing the number of hidden layers $L$) or "wider" (by increasing the number of nodes per layer $N_H$). We tested this by doubling each of these shape parameters, while keeping the other the same, and rerunning the DeepCME training procedure. As results in Fig 4A and 4B indicate, changing the DNN shape parameters did not improve the accuracy of the estimates. However we found that when we increase the number of training trajectories (see Fig 4C), the accuracy of the estimates does improve and this improvement is quite substantial in some cases (e.g. sensitivity w.r.t. $\gamma$ for $\mathbb{E}(X_n(T))$ and $n = 10$).

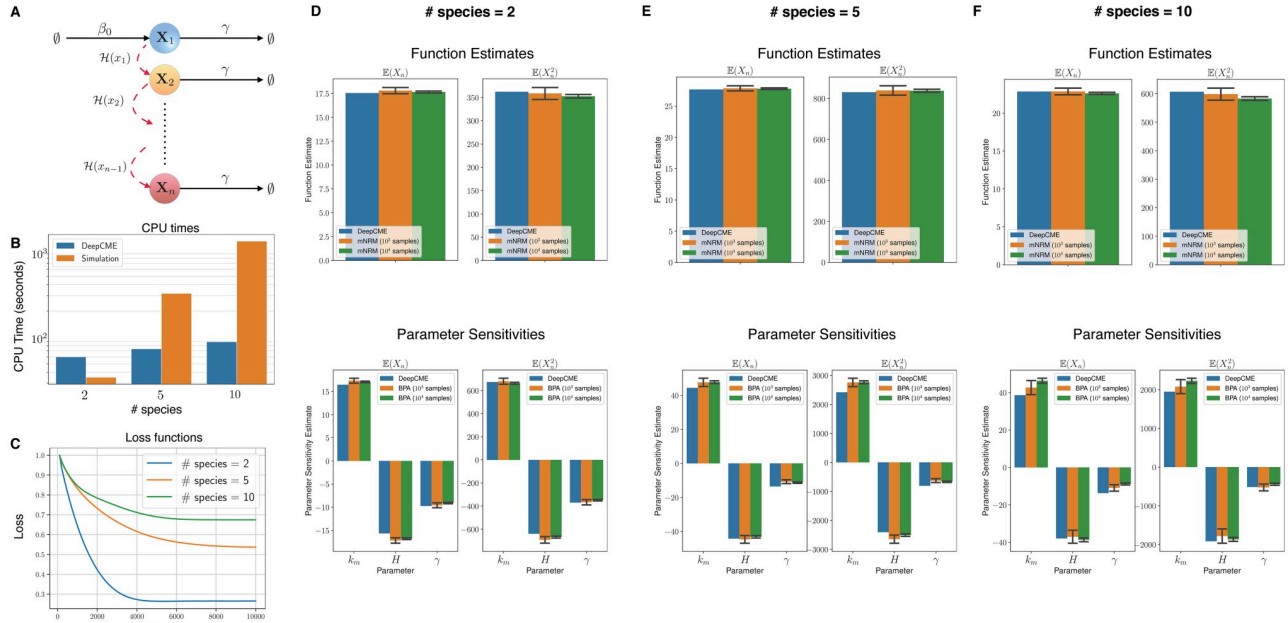

**Fig 5. Nonlinear signalling cascade.** (A) Depicts the network with *n* species and 2*n* reactions. The reactions shown with red dashed-arrow have propensities given by a nonlinear activating Hill function (32). Other reactions have mass-action kinetics. (B) The CPU times are shown for DeepCME for different values of *n* (denoted as # species), and for comparison the time needed by simulation based methods (mNRM for function estimates and BPA for parameter sensitivities) with 1000 trajectories is also indicated. (C) Plots the validation loss function w.r.t. training steps for various *n* values. Panels (D-F) show estimates for the function values ($\mathbb{E}(X_n(T))$ and $\mathbb{E}(X_n^2(T))$) at T = 1 and the parameter sensitivities (only the significant sensitivities are shown). The estimates with simulation based methods are shown as 95% confidence intervals.

### 5.3 Nonlinear signalling cascade

We now consider a variant of the network in the previous example where the catalytic production of species $\mathbf{X}_{i+1}$ by species $\mathbf{X}_i$ is non-linear (see Fig 5A) and is given by a activating Hill propensity with a basal rate

$$\mathcal{H}(x) = b + \frac{k_m x_i^H}{k_0 + x_i^H} \tag{32}$$

where $b = 1$, $k_m = 100$, $k_0 = 10$ and $H = 1$. Other reactions have mass-action kinetics as in the previous example, with the same rate constants $\beta_0 = 10$ and $\gamma = 1$.

For number of species $n = 2, 5, 10$, we apply DeepCME to this reaction network by training the neural network for 10′000 SGD iterations. Then we compute the moment estimates (31) and their sensitivities to all the model parameters. These quantities are also estimated with simulation-based methods (mNRM and BPA) with 1000 samples, and unlike previous examples we cannot compute these quantities exactly due to nonlinear propensities. In the plots shown in Fig 5D, 5E and 5F, we compare the estimates from DeepCME and simulation-based approaches for various values of *n*. Observe that DeepCME is reasonably accurate in estimating the moments and their parametric sensitivities for all values of *n*. The success of the training process is shown by the validation loss function profiles in Fig 5C. Note that these loss functions increase monotonically with *n* and this is consistent with the observation that errors in DeepCME-estimated quantities increase with *n* (see Fig 5D, 5E and 5F). The CPU times for DeepCME and simulation-based methods are displayed in Fig 5B, and as in the previous examples they show sub-linear growth w.r.t. *n* for the former but linear growth for the latter.

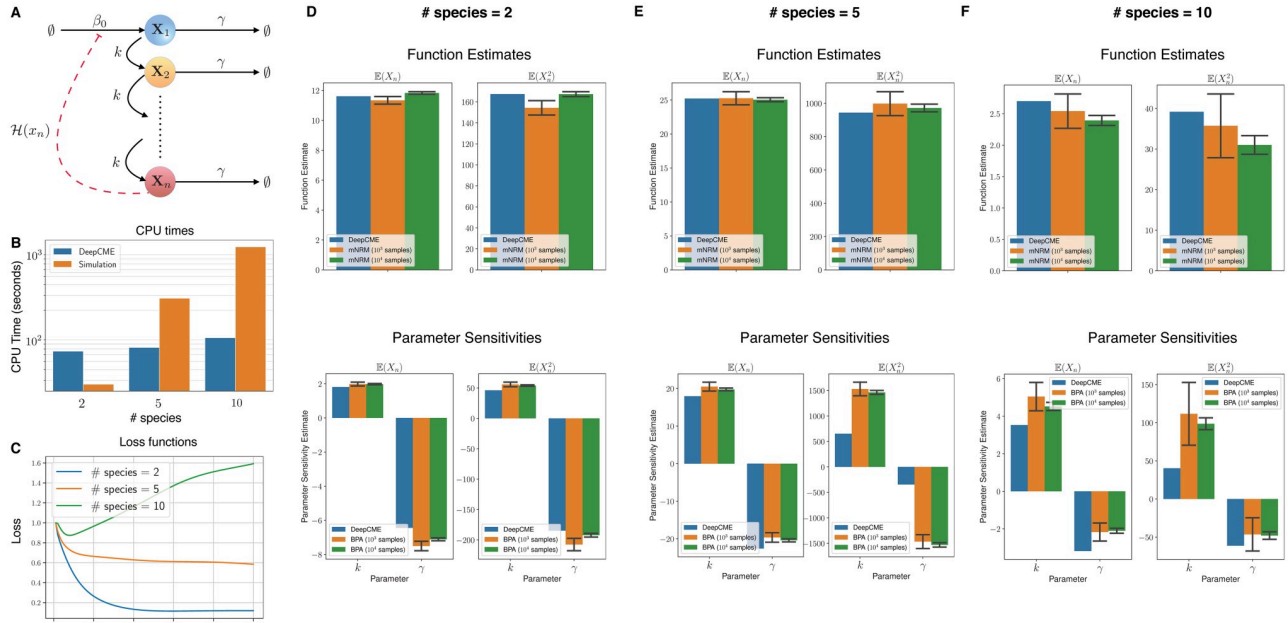

**Fig 6. Linear signalling cascade with feedback.** (A) Depicts the network with *n* species and 2*n* reactions. The reaction shown with red dashed-arrow has propensity given by a nonlinear repressing Hill function (33). All other reactions have mass-action kinetics. (B) The CPU times are shown for DeepCME for different values of *n* (denoted as # species), and for comparison the time needed by simulation based methods (mNRM for function estimates and BPA for parameter sensitivities) with 1000 trajectories is also indicated. (C) Plots the validation loss function w.r.t. training steps for various *n* values. Panels (D-F) show estimates for the function values ($\mathbb{E}(X_n(T))$ and $\mathbb{E}(X_n^2(T))$) at T = 1 and the parameter sensitivities (only the significant sensitivities are shown). The estimates with simulation based methods are shown as 95% confidence intervals.

## 5.4 Linear signalling cascade with feedback

Lastly we consider another variant of the network in the second example where there is negative feedback in the production of $\mathbf{X}_1$ from species $\mathbf{X}_n$ (see Fig 6A) which is given by a repressing Hill function with a basal rate

$$\mathcal{H}(x) = b + \frac{k_m}{k_0 + x_n^H}, \tag{33}$$

where $b = 1$, $k_m = 100$, $k_0 = 10$ and $H = 1$. Other reactions have mass-action kinetics as in the second example, with the same rate constants $k = 5$ and $\gamma = 1$. Due to the presence of feedback, oscillations can arise in the dynamics and to better represent this temporal dependence of the policy map we encode it with $N_T = 5$ identical DNNs (see Remark 4.3).

For number of species $n = 2, 5, 10$, we apply DeepCME to this reaction network by training the neural network for 10′000 SGD iterations. Then we compute the moment estimates (31) and their sensitivities to all the model parameters, and we also estimate these quantities with simulation-based methods (mNRM and BPA) using 1000 samples. In the plots shown in Fig 6D, 6E and 6F), we compare the estimates from both these approaches for various values of *n*. Observe that DeepCME is quite accurate in estimating the moments for $n = 2, 5$ and the parametric sensitivities for only $n = 2$. For $n = 5, 10$ only the sensitivities for $\mathbb{E}(X_n(T))$ are accurate but the sensitivities for $\mathbb{E}(X_n^2(T))$ are not accurate with our neural network architecture. The validation loss function trajectories are shown in Fig 6C. The CPU times for DeepCME and simulation-based methods are plotted in Fig 6B, and they show a similar growth pattern as our earlier examples.

## 6 Conclusion

Over the past couple of decades, stochastic reaction network models have become increasingly popular as a modelling paradigm for noisy intracellular processes. Many consequential biological studies have experimentally highlighted the random dynamical fluctuations within living cells, and have employed such stochastic models to quantify the effects of this randomness in shaping the phenotype at both the population and the single-cell levels [40]. As experimental technologies continue to improve at a rapid pace, it is urgent to develop computational tools that are able to bring larger and more realistic systems within the scope of stochastic modelling and analysis.

The central object of interest in stochastic reaction network models is a high-dimensional system of linear ODEs, called the *Chemical Master Equation* (CME). Numerical solutions to the CME are difficult to obtain and commonly used simulation-based schemes to estimate the solutions often require an inordinate amount of computational time, even for moderately-sized networks. Inspired by the recent success of machine learning approaches in solving high-dimensional PDEs [13], our goal in this paper is to devise a similar strategy, based on deep reinforcement learning to numerically estimate solutions to CMEs. We develop such a method, called *DeepCME*, and we illustrate it with a number of examples. The neural network we train in DeepCME provides estimates for expectations based on the CME solution and in principle it also provides estimates for the sensitivities of these expectations w.r.t. all the model parameters without any extra effort. Such parametric sensitivities are important for many applications, such as evaluating a network's robustness properties [41] or identifying its critical components [42], but they are even more difficult to estimate than solutions to the CME [23–31].

Our work opens up several directions for future research. The machine-learning based computational framework and the mathematical formulation which we provide allows one to deploy and transfer strong ML methodologies to the quantitative analysis and to data assimilation into complex CRNs. The present, basic approach can be improved/extended in a number of ways.

Firstly, it needs to be investigated how the architecture of the neural network can be optimally selected, for improved convergence of the training process, based on the CRN model. Overparametrized neural network architectures may be regularised by adding suitable weight-bias penalties in the loss-function. The resulting improved convergence will increase the accuracy of the estimates provided by DeepCME, especially for the parameter sensitivities, and reduce the number of trajectories needed for the neural network training.

Secondly, although the presently proposed framework requires relatively few 'exact' stochastic simulations of the dynamics, it could nevertheless be computationally prohibitive for many large biological networks, especially if they are multiscale in nature [43, 44]. It might be possible to improve efficiency by replacing exact simulations with $\tau$-leaping simulations [20], multi-level schemes [21] or with simulations based on reduced models for multiscale networks [16, 43–45]. Incorporating such approaches for generating training trajectories would make our approach computationally feasible for much larger networks than those considered here. In particular, *multi-level simulation schemes* which are based on coupling techniques [21] would allow one to construct a lower variance estimator for the loss function (29). This could in turn benefit the accuracy and the convergence of the training process (see, e.g. [46] for the development of multilevel DNN training algorithms, albeit in another class of applications). In the context of multiscale networks, identifying the appropriate copy-number scalings that give rise to reduced models with simpler dynamics is a highly specialised task requiring careful theoretical analysis [16]. However our approach can be extended to "learn" these scaling factors

during the training process by including them as trainable subnetworks into the ML feature space and employing them to scale the state-vectors in the input layer of the DNNs (see Fig 1). It is quite possible that incorporating these scaling factors would enhance the expressivity of the DNN.

Thirdly, the parameter sensitivities that we compute in our method could be employed in an 'outer' gradient descent method with the purpose of inferring model parameters by matching the computed statistics of CME solution with experimental data [47].

On the theoretical front, greater mathematical effort is required to understand how deep reinforcement-learning approaches can help in circumventing the curse of dimensionality inherent to CMEs. Alternative training approaches, such as *Generative Adversarial Nets* (GANs), may also be suitable for acceleration of the training process (see, e.g., [48]).

Finally, the architecture of the DNNs may include feature spaces comprising *parametric dictionaries of motifs*, which are adjusted during training to the reaction rates and to the kinetics of the CRN under consideration.

## Author Contributions

**Conceptualization:** Ankit Gupta, Christoph Schwab, Mustafa Khammash.

**Formal analysis:** Ankit Gupta.

**Funding acquisition:** Mustafa Khammash.

**Methodology:** Ankit Gupta, Christoph Schwab.

**Resources:** Mustafa Khammash.

**Software:** Ankit Gupta.

**Validation:** Ankit Gupta.

**Visualization:** Ankit Gupta.

**Writing – original draft:** Ankit Gupta.

**Writing – review & editing:** Ankit Gupta, Christoph Schwab, Mustafa Khammash.

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
