## [Decision Letter · Decision Letter 0]

31 Jul 2021

Dear Prof. Khammash,

Thank you very much for submitting your manuscript "DeepCME: A deep learning framework for solving the Chemical Master Equation" for consideration at PLOS Computational Biology. As with all papers reviewed by the journal, your manuscript was reviewed by members of the editorial board and by several independent reviewers. The reviewers appreciated the attention to an important topic. Based on the reviews, we are likely to accept this manuscript for publication, providing that you modify the manuscript according to the review recommendations.

Sincerely,

James R. Faeder

Associate Editor

PLOS Computational Biology

Daniel Beard

Deputy Editor

PLOS Computational Biology

[LINK]

Reviewer's Responses to Questions

**Comments to the Authors:**

Reviewer #1: review is uploaded as attachment

Reviewer #2: At a high level, the authors are using machine learning techniques (basic feedforward neural networks) to solve for (time dependent) expectations of stochastically modeled reaction networks. I think that this paper has the potential to be very good. It is the rare paper I review in which I think to myself "I wish I had done this!". Here are my main comments/critiques that I would like the authors to consider.

1. The title claims to solve the chemical master equation (Kolmogorov's forward equation). However, the paper is actually focused on solving for expectations. Now, I agree that a solution to the CME can come from using indicator functions. However, that does not really seem to be a natural application for this method (and there are no examples showing if the method is good for that).

2. A very important part of the paper is Theorem 3.3. More precisely: equation (19), which gives an equality that must hold (almost surely) for every *path*. In my view, the hear of the proof is that m(t) (unnumbered equation between (23) and (24)) is a local martingale. A few more words can be given here (or a reference) proving/saying that \\Delta \\hat V_k(s,X(s)) is adapted to the proper filtration and \\tilde R_k is a local martingale, thus.... Pointing to the proper reference will be helpful to readers not familiar with these things.

3. Page 10. Why was \\phi as given chosen? (this is minor)

4. This is my only major critique. I could not figure out exactly what the input to the NN was. This is explained on pages 10 and 11, but I did not understand it. Of course, I would really like to know how a path gets inputed since this is an absolutely key part of the method. I wonder if a simple example would be helpful here (I would suggest the birth death process of section 5.1).

5. Why were the NN's used so small (L = 2 and N_H = 4)?

6. The method does not seem viable at this stage for computing sensitivities. Perhaps soften your language to something along the lines of "we note that sensitivities can theoretically be computed as well via these methods" you could then give the reasoning as is. However, you could point out that it sometimes works and sometimes doesn't (since the examples are all over the map), and then point to this as future research.

Reviewer #3: The DeepCME is a novel deep learning framework for numerical solution of the chemical master equation (CME). The authors reformulate the problem of obtaining expectation of specific functions of state space and their sensitives using the Kolmogorov's backward equation (theorem 3.3). Using this reformulation they construct appropriate loss functions to train deep neural networks using a reinforcement framework using relatively small number of stochastic simulations (SSA). They demonstrate the power and utility of the their method using a series of examples. This is a timely, well written and important study. I have the following specific comments:

- Could the authors provide some insight the advantage of the specific formulation compared for example to forward equation in solving CME using deep neural networks?

- At the beginning of section 5 the author's describe specific choice of a large number of hyper parameters (number of layer's, nodes, etc). Could the authors explore the significant of at least some of these choices in an example on the performance?

- The author's show the CPU time required for direct SSA vs DeepCME. Could the author's also make comparison in term's of number of simulations used in the training and how does that change with n and the quality if the results?

**Have the authors made all data and (if applicable) computational code underlying the findings in their manuscript fully available?**

Reviewer #1: Yes

Reviewer #2: Yes

Reviewer #3: Yes

PLOS authors have the option to publish the peer review history of their article (what does this mean?). If published, this will include your full peer review and any attached files.

Reviewer #1: No

Reviewer #2: No

Reviewer #3: **Yes: **Vahid Shahrezaei

Figure Files:

Data Requirements:

Reproducibility:

References:

---

## [Decision Letter · Decision Letter 1]

18 Oct 2021

Dear Prof. Khammash,

Thank you very much for submitting your manuscript "DeepCME: A deep learning framework for computing solution statistics of the Chemical Master Equation" for consideration at PLOS Computational Biology. As with all papers reviewed by the journal, your manuscript was reviewed by members of the editorial board and by several independent reviewers. The reviewers appreciated the attention to an important topic. Based on the reviews, we are likely to accept this manuscript for publication, providing that you modify the manuscript according to the review recommendations. In particular, Reviewer 2 is asking for clarification on the exact form of input to the neural net, which would be needed to replicate the method.

Sincerely,

James R. Faeder

Associate Editor

PLOS Computational Biology

Daniel Beard

Deputy Editor

PLOS Computational Biology

[LINK]

Reviewer's Responses to Questions

**Comments to the Authors:**

Reviewer #1: The authors have sufficiently addressed my comments.

Reviewer #2: The manuscript has improved. However, one of my main worries about the manuscript (and apparently this worry was also shared by Reviewer #1) remains unresolved. In particular, the inputs to the NN are still quite unclear to me. Let me be very explicit: I have no idea what is going on with the lambda's and phi's of what is now section 4.1. The authors describe them as "temporal features", but that doesn't mean anything to me. A trajectory is simply a listing of states and times. The authors need to explain how a trajectory is inputed into the NN. If there is a mapping to the lambda's and phi's somehow, then so be it, but please tell us clearly what that mapping is. At this point I would not be able to implement this method as the whole portion on "temporal features" is mysterious.

Here are some smaller comments:

1. In the abstract it again says “The goal… estimating solutions of high-dimensional CMEs…” this is not really what it is doing. The next line is more accurate. Maybe merge and fix? (Except the next line says "Arbitrarily chosen functions" which seems strange)

2. (Minor). Line 63. “Which for” to “for which”

3. Line 123, this is super small but in the definition of the Q matrix you seem to be assuming that each reaction vector is uniquely associated with a reaction. This is not always the case.

4. Line 143. “On which our deep learning approach depends on” — one too many “on”s

5. Same line. Which is same —> which is the same.

6. Line 250. Why is this line here: “The relation between Φ and its realisation R(Φ) as a map is not one-to-one.” I guess it’s true: there can be lots of choices of rho’s and T’s that give R. But is this important (i.e., I’m concerned I’m missing something)?

Reviewer #3: The authors have addressed all of my concerns in the revised manuscript.

**Have the authors made all data and (if applicable) computational code underlying the findings in their manuscript fully available?**

Reviewer #1: None

Reviewer #2: Yes

Reviewer #3: Yes

PLOS authors have the option to publish the peer review history of their article (what does this mean?). If published, this will include your full peer review and any attached files.

Reviewer #1: No

Reviewer #2: No

Reviewer #3: No

Figure Files:

Data Requirements:

Reproducibility:

References:

---

## [Editor Report · Decision Letter 2]

8 Nov 2021

Dear Prof. Khammash,

We are pleased to inform you that your manuscript 'DeepCME: A deep learning framework for computing solution statistics of the Chemical Master Equation' has been provisionally accepted for publication in PLOS Computational Biology.

Best regards,

James R. Faeder

Associate Editor

PLOS Computational Biology

Daniel Beard

Deputy Editor

PLOS Computational Biology

---

## [Editor Report · Acceptance letter]

22 Nov 2021

PCOMPBIOL-D-21-01072R2 

DeepCME: A deep learning framework for computing solution statistics of the Chemical Master Equation

Dear Dr Khammash,

I am pleased to inform you that your manuscript has been formally accepted for publication in PLOS Computational Biology. Your manuscript is now with our production department and you will be notified of the publication date in due course.

With kind regards,

Anita Estes
